# The Contribution of the Architect Pascuala Campos to the Implementation of a Gender Perspective in the Galician Context

**Ainoa Fernández Cruces** *, **Goreti Sousa** *, **Paulo Guerreiro and Mariana Correia**

Ci-ESG—Escola Superior Gallaecia, 4920-275 Viana do Castelo, Portugal; pauloguerreiro@esg.pt (P.G.); marianacorreia@esg.pt (M.C.)

* Correspondence: ainoafdezcruces@gmail.com (A.F.C.); goretisousa@esg.pt (G.S.)

**Abstract:** The incorporation of women in society, as active professionals, was probably one of the most important parameters of modernity in the last century. Until the beginning of the twentieth century, women who entered the world of architecture were, generally, assigned to the design of domestic interiors. Thus, they were always in the background, which contributed to the concealment of the female gender perspective in architecture and an incomplete vision of its history. The general purpose of this article is to address the implicit problematic of the female contribution to architecture, through a theoretical reflection that aims at recognizing the relevant impact of Pascuala Campos's work to the discipline in Galicia, Spain. The Spanish social and architectonic contexts, as well as the biography of Pascuala Campos, are analyzed to better understand her theoretical and architectonic production. The analysis combines data from different sources, mainly documental research, interviews, and architectonic surveys. The basic principles stressed in the theoretical production of Pascuala Campos are thus identified and served as analytic categories for the survey of the Combarro Urban Intervention. These results allowed the identification of concepts and projected guidelines interpreted as gender perspective-oriented.

**Keywords:** architecture and urban planning; gender perspective; Pascuala Campos

## 1. Introduction

Pascuala Campos de Michelena was a pioneer in several aspects of Galician architecture. She was the first female architect recognized for her professional work, the first to be assigned a Chair in Architecture and Urban Design at a Spanish university and also a pioneer in her civic work battling for women rights. Her life and work are the focus of the present research paper, trying to understand the way she managed to succeed and to inspire other generations of women who followed her example and devoted their life to architecture, despite a background of profound discrimination towards women.

Between 1970 and 2012, several of the projects that she developed, either by herself or in partnership, are referenced in several publications (Baldellou 1971; Climent Ortiz 1976; 2C. Construcción de la ciudad 1977; Almuíña Díaz 1977; Bru and Mateo 1984; Tzonis and Lefaivre 1985; On Diseño 1987; AV 1993; Xunta de Galicia 1997; Lasaosa 2000; Mesía López 2010).

Recently, her life and work have been the object of attention. In 2000, *Andaina*, a feminist Galician magazine published the interview conducted by Berta Rey Wonenburger to the architect. But the first paper to address her career was published in 1993 in the book *Galicia-Arte*, edited by Rodríguez Iglesias (1993). In 2011, she was mentioned in the report on *Las mujeres arquitectas de Galicia: su papel en la profesión (el ejercicio de la arquitectura en Galicia desde una perspectiva de género)*. Amparo Casares Gallego is probably the researcher who published more papers about Pascuala Campos on her own (Casares Gallego 2002, 2004) or in collaboration with Casares Gallego and García (2013).

The present research paper picks up an analysis developed for the Master's dissertation "*Women Architects and their contribution to the discipline: Lina Bo Bardi and Pascuala Campos*" (Fernández Cruces 2019), which addressed the social and biographical background of the two architects and the interpretation of their theoretical production and architectonic and/or urban planning work. A research strategy was developed by interconnecting data from multiple sources, mainly documental analysis and an architectonic survey that was also used in the present paper.

Bearing this background in mind, the present research paper tries to add to the discussion about Pascuala Campos's contribution to Galician architecture by interpreting these established facts about Pascuala Campos's life and work against her own theoretical frame regarding gender and architecture and their materialization in architectonic and urban forms throughout her professional work.

Therefore, the concepts of sex, gender and gender perspective set the theoretical background of this research paper.

Gender is understood as a social construction which assigns different roles according to sex. Furthermore, it seems clear how this social construction of gender is linked to imposed physical and symbolic spaces (Osborne and Molina Petit 2008).

Feminism, or feminisms to be more precise, is defined as a philosophical and political struggle (Muxí Martínez 2018) with multiple ideological and philosophical lines, although all united by their commitment to the emancipation of women (Hoffman 2002). Therefore, it has a different understanding from a gender perspective, since the last does not address women exclusively (Muxí Martínez 2018), but rather aims at intersectionality.

For the aim of this paper, three interconnected perspectives concerning gender architecture are adressed: the personal and professional biography of Pascuala Campos, in order to determine the conditions through which she had access to the profession in a period where female architects in Galicia were scarce; and her integration in the historical narrative of Galician architecture is also analyzed to finally examine the way Pascuala Campos envisions and designs spaces, concentrating on her theoretical production and surveying her projects, developed in the time span covered in this Special Issue.

## 2. Pascuala Campos: Access to the Profession

Pascuala Campos was born in Sabiote, Jaén, in 1938, where she lived until 1939 when, in the aftermath of the Spanish Civil War, her father, a former Captain of the Republican Army was arrested and exiled, forcing the family to move to Navas de San Juan. In 1941, because of her father's work and the rising of fascist repression, the family was, once again, forced to move, this time to Trebujena, Cádiz. Later, in 1948, after the third arrest of her father, the whole family moved to Arrollo de Ojanco, Jaen; and finally, in 1955, to Beas de Segura, within the same province.

This was a particularly decisive moment for Spanish architecture, which naturally reacted to the socio-cultural context of ascension to power of a nationalist, totalitarian regime. By the time Pascuala was born, almost at the end of the Civil War, Franco's regime repressed the modern approach that was lead by GATEPAC in the 1920s, now branded as *Jewish and stateless*. Few were those who were able to continue to work with the same architectonic orientation they had pursued before the war. Only in the mid-1940s, Spain, still under Franco's dictatorship, started to slowly open to the exterior, making it possible for Spanish architects to have a more accurate sense of what was going on in the rest of the world, enabling some proposals to resume the previous reflections, and the preparation of the architectonic renovation of the next decade (Sambricio 1994).

In 1956, Pascuala moved to Madrid to prepare her admission to *Escuela Técnica Superior de Arquitectura de Madrid*, which finally happened in 1960. In 1965, due to the rising political pressure in Madrid, Pascuala, who, while in Madrid, supported student organizations for political vindication, such as the *Federación Universitaria Democrática Española* (FUDE), moved to *Escola Tècnica Superior d'Arquitectura de Barcelona* (ETSAB), so she could complete her degree. Pascuala Campos was the third

woman to accomplish a degree in architecture at ETSAB, in 1966. Two years later, she obtained her PhD at the same university.

That same year (1968), following her marriage with César Portela two years before, she moved to Pontevedra, in Galicia, where the couple started their architecture studio.

Particularly in Galicia, the 1950s were a period of great changes that coincided with the arrival of José Bar-Bóo and Andrés Fernández-Albalat and their effort to introduce in Galicia the new architectonic principles they were thought in Madrid (Baldellou 1971).

Galician architecture had been dominated, for almost one decade, by José Bar-Bóo and Andrés Fernández-Albalat, when, from 1965 onwards, a second wave of young architects arrived with fresh, new ideas: Manuel Gallego, Javier Suances, Rafael Baltar Tojo, José António Bartolomé Argüelles and, of course, the couple César Portela and Pascuala Campos (Baldellou 1971). Together, they started a coherent front towards common goals such as the renovation of the professional organizations and architectonic interventions that both showed more critical sense and consolidated the social status of the architect.

José Bar-Bóo and Andrés Fernández-Albalat had already shown a tendency to intersect modernity with Galician architectonic tradition, but this new generation showed a higher sensitivity to local building culture, following the example of Stirling. They engrossed what is called critical regionalism.

As Tzonis and Lefaivre (1985) described it, critical regionalism was an attitude towards the collective memory and the spirit of the place as a way to interpret human relations, which implied a renovation of the architectonic language based on a profound knowledge of the local building culture.

Between the end of the 1960s and in the 1970s, the studio of Pascuala Campos and César Portela developed several architectonic projects, some of which were widely discussed, and thus attracted attention to their work. For instance, the *Gypsy settlement of Campañó*, 1970–74, and the *Market and Fish Market of Bueu* (with the collaboration of Luis López de Castro), 1971, both in Pontevedra, were part of the travelling exhibition *Architecture and Rationalism. Aldo Rossi + 21 Spanish Architects* (1975).

But this growing attention towards their work was very much fuelled by the strong personality of César Portela, and Pascuala Campos was left somewhat in his shadow. This kind of discrimination was a prevailing attitude in those years. Pascuala Campos herself recognized this in her interview with Berta Rey Wonenburger, when she mentioned that, in her promotion, there were only five women, all married architects, and she worked in partnership with them. All the prominence went to the male partner. In her words, they fought for it and they took efforts in rendering their wives' work invisible (Rey Wonenburger 2000).

The democratic transition that Spain went through in the late 1970s and early 1980s brought forward new opportunities for architects, and Pascuala Campos took full advantage of these opportunities. She first started a new architecture studio with Ana Fernández Puentes (1980) and later, in 1982, she set up her independent studio in Pontevedra, where she developed several projects, among which were the *Plan for the urban intervention in Combarro* (1984), the *Fishery Arts School of Isla de Arousa* (1990), the *Project for a Funeral home in Bueu*, and the *Intervention in the Public Park of Bueu* (2000), in collaboration with Amparo Casares Gallego.

At the beginning of the 1980s, she initiated a parallel career as a professor in the Department of Architectural and Urban Design at the *Escuela Técnica Superior de Arquitectura* in A Coruña, at the time integrated in the Santiago de Compostela University (1982), where she was appointed director of the design department in 1986. In 1995, the *Escuela Técnica Superior de Arquitectura* was already part of the A Coruña University, and Pascuala Campos achieved her Chair in Architecture and Urban Design, becoming the first woman ever appointed to this position in a Spanish university.

With the social changes that followed the end of Franco's dictatorship, another relevant aspect of Pascuala Campos's work started. Her feminist concerns made her, in collaboration with Ana Fernández, María del Carmen Dios, and Aurora Domínguez, among others, enrol in several feminists' associations, such as the Pontevedra Group (1976). This group was linked to the Galician Women's Association (*Associación Gallega da Muller*, AGM). But the association was devastated by discrepancies between

those known as the independent militants and the double militants, who were accused by the first group to have a parallel political agenda. This rupture led to the exit of the independents, where some of them, including Pascuala Campos, started in 1978 the Galician Independent Feminists (*Feministas Independentes Galegas*, FIGA). Two years later, their example was crucial in the formal rupture of the independent feminists that took place in the feminist meeting in Granada.

FIGA organized large independent feminist meetings, but mainly promoted smaller meetings and debate groups for a more direct contact with Galician women, in order to discuss issues concerning sexuality, such as the use of contraceptives and abortion.

In Pascuala Campos's writings, it seems clear how these sexual claims are the pillars on which she forged her own understanding of what it means to be a woman. She describes how centuries of legal and sexual control over women made them interiorize atitudes of comprehension, attention, affection, and sexual disponibility towards their male counterparts. It also drove them to tend to the wellbeing of others as a responsibility. This effort, not being paid or officially recognized, was rendered "natural", and thus invisible (Campos de Michelena 1996, p. 26).

Furthermore, her claims as a feminist activist were determinant to her work as an architect, since, as she very well describes, she enters her professional career "with the feminist understanding of the world" (Campos de Michelena 1999).

For her, despite the patriarchal society that removes the female collective from the decisions concerning the organization of space (Campos de Michelena 1996), "the concept of space is deeply linked to the feminine" (Campos de Michelena 1995a, p. 229), for the feminine body is the only one that can deeply understand the concept of being inhabited (Campos de Michelena 2000). Therefore, one can infer that only they are able to achieve an architecture that highlights the subtle complexity of human relations (Campos de Michelena 1995b).

In the next decade, she continued her action as feminist activist, participating in the creation of the Galician Independent Feminists Group (*Grupo de Feministas Independientes de Galicia*, FEIN) in 1999, following the dissolution of FIGA in 1984, while her participation in several initiatives concerning the discussion of gender architecture and urban planning grew exponentially.

Her civic work and contribution to the Galician feminist cause is constant until today. Despite her jubilation as a professor, after more than twenty years teaching, Pascuala Campos still participates in debates and seminars about gender and architecture and was appointed to the Galician Culture Council Equality Commission from 2007 to 2016.

## 3. Pascuala Campos in the Galician History of Architecture Narrative

Like many other social constructions, the history of architecture is based on established narratives, that were, historically, mainly shaped by men. Therefore, women's perspective was silenced in a way that can compromise the hypothetical neutrality of the historical discourse. This conscience aroused the necessity of a profound revision of the official narrative, to establish female referents in architecture and urban planning. Regarding the Galician history of architecture, Pascuala Campos is one of those new references.

However, although from a very early stage in her career her work has been the object of attention and discussion, one can observe a great evolution in the way she has been depicted.

In the Introduction of this paper, a state of the art about her life and work was presented. Analyzing those texts, it becomes clear how the first references to Pascuala Campos can be found in publications which present projects developed in partnership with her husband. Although she is mentioned, it is clear how all the praise goes to César Portela. It was a very natural attitude in a time when women architects were still a rarity and those who worked had a partnership with their husband, who took all the credit and merit of their joint work.

Paradigmatic is the article published in 1971 by Miguel A. Baldellou, in which her name is cited with other architects, all male, referencing those as a new and promising generation of architects in Galicia. Pascuala Campos is mentioned as a member of the partnership Portela–Campos. But when

Baldellou, further along the text, analyzes their influence in Galician architecture, Pascuala Campos is completely forgotten: "The personality and strength of César Portela has been working as a catalyst of the better Suances, of the more combative Bar and possibly of the more dynamic future *Colegio de Galicia*" (Baldellou 1971, p. 49). It is an attitude that fits the mentality of those times, when the man was always seen as the public face of any partnership.

The year of 1985 seems to be a turning point in this andocentric perspective where the wife is mentioned as a mere collaborator in the projects of the husband and rendered invisible to public recognition. In what concerns Pascuala Campos's work, two texts are exemplary of this changing perception.

The first, with the title *Young Spanish architecture*, published once more several projects developed in partnership between Pascuala Campos and her husband, but they appear under the scope of César Portela's work, with only a small reference to her collaboration, as if her reference was merely a formality (Campo Baeza and Poisay 1985).

The second text is published that same year by Alex Tzonis and Liane Lafaivre, in which one can already observe a different perspective. The authors presented the critical regionalist movement in Spain choosing for each region one representative project, and Pascuala Campos's intervention in Combarro was chosen as a reference for Galicia. Furthermore, they recognized Pascula Campos's gender perspective as one key characteristic of her work.

Baldellou himself, in another work published in 1995, seems to realize the error of his previous omission. Twenty-five years later, he recognizes that Pascuala Campos had an active role in the projects developed in partnership with César Portela, acknowledging the possibility of mutual influence, therefore admiting, possibly with some regret, that her personality may have been, in those years, excessively relegated to the background because of her female condition. Furthermore, he admits how she resisted in her convictions until the opportunity of expressing them with great strenght in her later work, and ended stating that: "We should be aware of what this voice may say in the future" (Baldellou 1995, p. 33).

By this time, her urban intervention project for Combarro had already been included in the exhibition *Galician Institutional Architecture* in 1991. This exhibition displayed 41 projects, but only two had female authorship. Pascuala Campos's was one of them and the other was a project of Chus Blanco in partnership with Fernando Blanco. Only one more woman is mentioned in the catalogue of the exhibition, Pilar Díes Vázquez, a collaborator of Alberto Noguerol del Rio.

A more solid valorization of her life and work was already on the way. Another hinge moment is the beginning of the twenty-first century when Pascuala Campos is, finally, recognized as a pioneer in Galician architecture, mainly through her distinctive gender perspective scope.

Her professional work has also deserved some recognition. In 1995, her proposal for the *Fishery Arts School of Isla de Arousa* (1990) won a national award and was selected to be exhibited in the exhibition *Women builds. Construction from Within* that showed a selection of 35 buildings by Spanish female architects. And in 2000, she won, with Amparo Casares Gallego, the outstanding ACCÉSIT prize for their project of a *Funeral Home in Bueu* and the *Intervention in the Public Park of Bueu*. Subsequently, in 2002, they received an honorary prize in the *International Architecture, Urban Development, and Sustainable Housing Contest*.

## 4. Pascuala Campos: Construction of Space

### 4.1. *Pascuala Campos's Conceptualization of Space and Gender*

Throughout her academic career, Pascuala Campos devoted her research to the necessities and consequences of spatial organization in the construction of identities, specifically focusing on gender identity. Her work is very vibrant while defending her personal view on architecture as a social construction determined by what she calls the microphysics of power, in which built space is a consequence of social, economic, cultural, psychological, and emotional circumstances (Campos de

Michelena 1999). Therefore, space from the public to the more intimate, cannot be seen as neutral since it reacts to predetermined conceptualizations of female/male (Campos de Michelena 1995a).

To Pascuala Campos, woman to woman relationships are tinged with the concept of fusion, while the man to man relationships are established from the concept of segregation (1999).

The problem arises when one realizes that, historically, the decisions concerning the organization of space relied exclusively on the male collective, according to their embedded values and ideologies. This resulted in high hierarchical spaces conceived from a perspective of power, which she ties with masculine identity (Campos de Michelena 1995a). The patriarchal structure of traditional societies lead to spaces hierarchically organized based on power, such as was already noted by de Beauvoir (1949), the most significant author on feminism of the twentieth century.

The female collective was excluded from the main decisions concerning public space. Therefore, everyday life was also rendered invisible and forgotten within the city (Campos de Michelena 1996).

This invisibility concerns also private spaces, which obey to rules imposed by those that occupy the public sphere: "The house was, and is, still, the place for women, but it is the private space of men" (Campos de Michelena 1999, p. 22).

The alternative depends on the female collective being allowed to participate in the decisions concerning the organization of space, but also on the reconnaissance that gender and sex are key concepts in understanding space and that everyday life may be a source for architectural forms (Campos de Michelena 1999). These new ways of understanding the city must rely on spaces where "the colecctive is an alternative to the public/private" (Campos de Michelena 2000, p. 357).

In her claim for a more active role of women in the decisions concerning space, one can read a valorization of intuition, that she defines as knowledge guided by feeling (Campos de Michelena 1995a). It was a belief aligned with those of Franck (1989), who associates masculinity with the denial of connection, with the objectivity that comes from rational detachment from the object. Femininity, on the other hand, is associated with self-identification through relationships, which infers subjectivity and emotion. These characteristics affect, from Franck's (1989) perspective, the method of analysis of reality and the construction of knowledge.

In architecture, these differences deeply affect the method of design, which by taking into consideration the emotional and the subjective, help strengthen the architect/client bond. In this sense, women tend towards spaces designed to highlight connections, to reduce differences between public/private, valuing flexibility and complexity (Franck 1989).

The importance Pascuala Campos recognizes to intuition in the method of design is one of her most distinctive traces. She defends that "the architect is not an objective technician, but a woman or a man submerged on their own personal circumstances" (Campos de Michelena 1999, p. 28).

In her writings, she is very critical of one particular idea of modernity, in which architecture is rendered abstract and decontextualized. "The design process is determined by categories imposed from outside of the person that designs in such a way that the act of disigning is reduced to a repetition of criteria where their own identity is annulled" (Campos de Michelena 1995b, p. 266). On the other hand, she believes that perceiving a space and responding to it, and its specific problems depends on being able to transgress self-identity (Campos de Michelena 1995a).

This new way of understanding architecture relies on the interconnection of physical and non-physical elements, such as feelings and traditions, among others (Campos de Michelena 1999).

These ideas (Figure 1) lead her to a defence of the architectonic process of design as an overlapping of several processes: first the reconnaissance; then the reconsideration of the needs and different ways of life implicated in a given program; followed by the recreation and finally the search for architectural form (Campos de Michelena 1999).

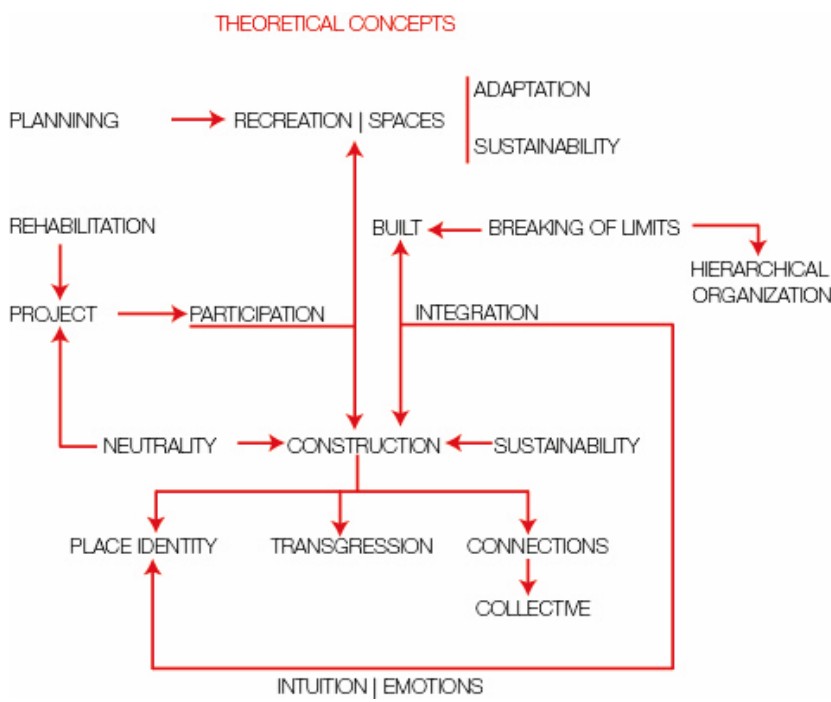

**Figure 1.** Synthesis of Pascuala Campos's theoretical production (Authorship: AFC).

The recreation of space is her way to find an alternative to those highly hierarchical spaces that she criticizes. The recreation of space "is about, on each moment, to search for beautiful spatial values that root us to the human, where the different lives of all sort of people is possible" (Campos de Michelena 1995a, p. 230), relying on designed spaces as spaces of communication.

The work of Pascuala Campos is "woven" through these concepts, showing a revolutionary attitude in a world in which women had few opportunities to build.

### 4.2. Pascuala Campos's Recreation of Space

These theoretical principles guided her professional career and can be identified in her interventions such as the *Plan and Proposals for urban intervention in Combarro* (1986) and the *Fishery Arts School of Isla de Arousa* (1988–1990). These were selected for a more detailed analysis due to their representativeness in the career of the architect and to their urban scope, which included different scales of spatial interventions, thus allowing the identification of a wide range of urban and architectural principles. Furthermore, those were developed in the time span covered in this Special Issue. Projects developed in collaboration with other architects were discarded, since the object of this analisys is to perceive how she materialized her own ideas into her projects.

### 4.2.1. Plan and Proposals for Urban Intervention in Combarro, Poio (1986)

Combarro, Poio, Pontevedra, is a small village that occupies a crescent-shaped peninsula between the beaches of Chouza and Padrón. Its history is linked to the sailors of Pontevedra and its relevance to the fishing harbor. The village identity depends on its nature, simultaneously linked to fishing and farming activities. This dual character is reflected in its urban morphology, with its buildings arranged along two streets, one towards the sea (San Roque) and the other to the interior (A Rúa). Another sign of this duality is the skyline of Combarro seen from the sea with its coastal wall and its granaries, and the urban space promotes the articulation between the two. The coastal wall defines the maritime profile of the settlement, interrupted by small roads perpendicular to the main street, which allow direct access to the sea. Its urban grid flexibly and creatively adapts to a rocky peninsula on a slope towards the sea, and forms small squares that open in such a way that all enable a sea view, almost all of them characterized by the distinctive presence of a cross or a fountain.

Although there are some one-story houses, the prevailing housing typology in Combarro is two-story, attached buildings, where the first floor is used as a wine house and warehouse and the second floor is left for domestic use. The living space is organized around the kitchen, with its fireplace and some cubicles aligned with a lateral corridor. One of the most characteristic elements of the village is the open balconies, supported by stone pillars arranged in narrow arched porticoes that were traditionally used for keeping the fishery instruments and similar articles, and nowadays form covered pathways.

These houses were customarily built with local stone, set in huge blocks and applied in walls, pavements, balconies, consoles, pillars, cornices, etc. These stone blocks were combined with wood, mainly in structural elements or carpentry, traditionally painted the same color as the family's boats.

Although Combarro was declared an Artistic Historic Site in 1972, at the beginning of the 1980s, architectural decay was already evident. The demographic growth of the 1970s with its necessity for adaptation of the traditional house resulted in the gradual introduction of foreign materials and construction techniques (Mesía López 2010). The decline was later enhanced by the decay of the local economy due to the abandonment of traditional fishing activities (Campos de Michelena 1985). Pascuala Campos highlighted another possible cause to this degradation, that is, the process of declaration of Combarro as an Artistic Historic Site did not involve the local community, and resulted in feelings of rejection and withdrawal of the new conditions that were imposed upon them (Campos de Michelena 1985).

In 1984, Pascuala Campos presented her plan and proposals for urban intervention in Combarro, developing, between 1986 and 1987, the Plan and Proposals for Restoration of Combarro's Traditional Granaries, the Special Plan for Interior Reform and the Catalogue of Buildings and Ensembles of the Municipality of Poyo. And later, in 1992–1993, the architect presented the Pilot Plan and Project for Urgent Intervention (Casares Gallego 2004).

The proposal as a whole includes two sets of recommendations, one for more immediate action and a second for long-term intervention.

The more urgent recommendations included:

1.　Maintenance interventions: pavement of the ramps that served as boat docks, replacement of the traditional stone pavement in some areas, restoration of the granaries, and consolidation of the coastal walls;
2.　Urban interventions: building a sewer system, redesigning the community's laundry and landfill, designing Rualeira Square (Figure 2), and a new path between Padrón´s beach and the north of the village.

Some political and administrative impasses delayed the implementation of these actions, and almost a decade passed until the pilot plan and the Combarro Urgent Actions Project were finally carried out. Meanwhile, the Special Plan for the Protection of the Historic Centre of Combarro was presented only in 1999 and was later approved in 2001 (Casares Gallego 2002).

Despite these delays, the plan presented by Pascuala Campos in 1984 (Figure 3) was largely discussed and finally gave her the public recognition that in previous works was concealed by her partnership with her husband. In 1984, Pascuala Campos published the *Typological Study (Pontevedra)*; in 1987, *On Design Magazine* published an article about the actions proposed for Combarro and in 1990 these proposals were included in the *Galicia Institutional Architecture exhibition*.

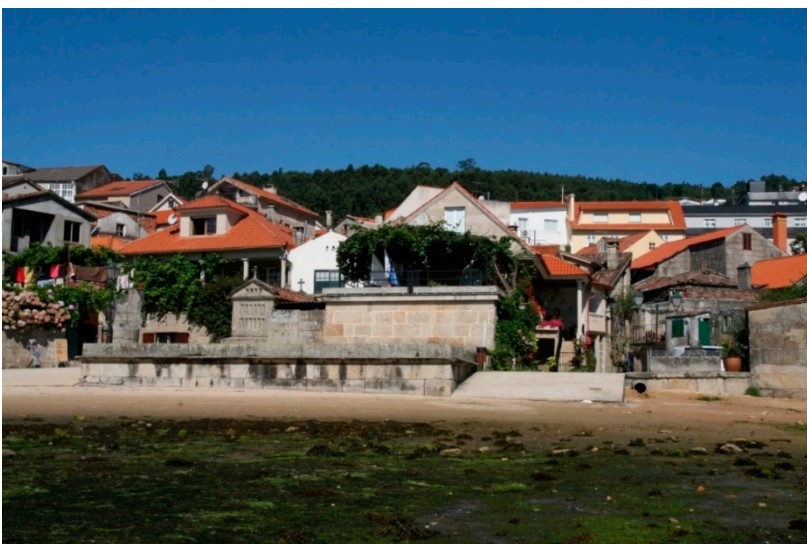

**Figure 2.** Photo of the Rualeira Square, showing the skyline of Combarro with the granaries and the coastal wall (Authorship: AFC).

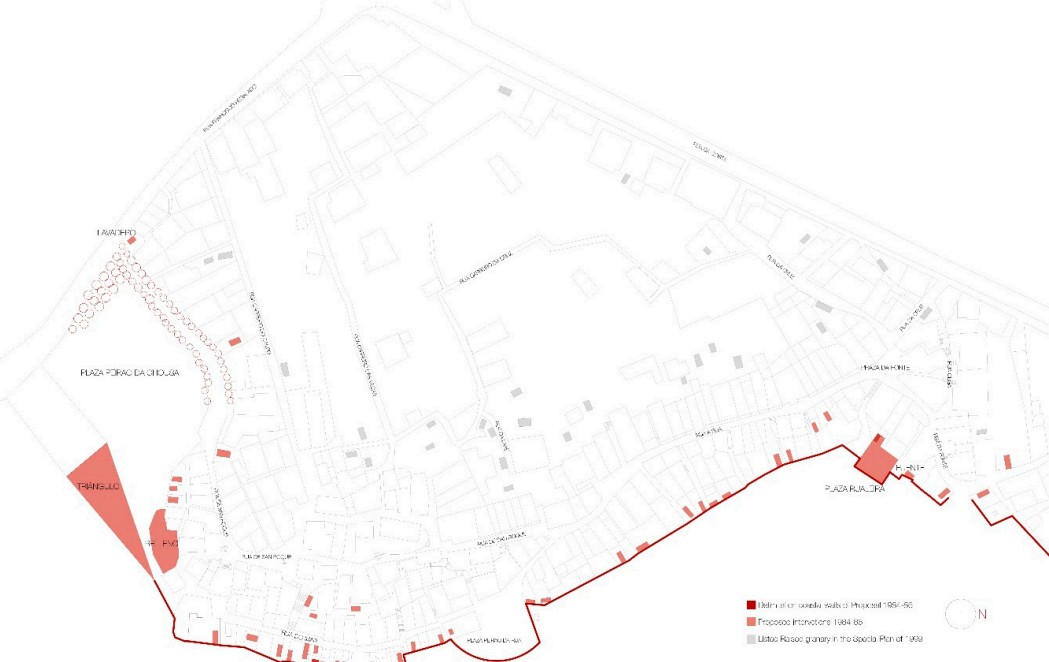

**Figure 3.** General plan of the Combarro Urban Intervention Proposal (Authorship: AFC).

These actions were designed to give dignity to the pre-existing space and to reaffirm the historical and cultural values of Combarro. Good examples of this attitude are the partial replacement of the traditional pavement, which allowed a considerable improvement of mobility, without compromising the local identity of a settlement built on top of a rock (Mesía López 2010), the restoration of the 38 granaries, or the new community's laundry.

This last intervention was very appraised by Tzonis and Lefaivre (1985) as one of the most interesting interventions within the Galician critical regionalist movement. The new community laundry was designed with two levels: the first was the covered area, defined by stone pillars where the wash tank was placed (Figure 4), and the second was a platform with no enclosing walls and a plastic roof where clothes are exposed to the air and the sun in order to dry (Figure 5).

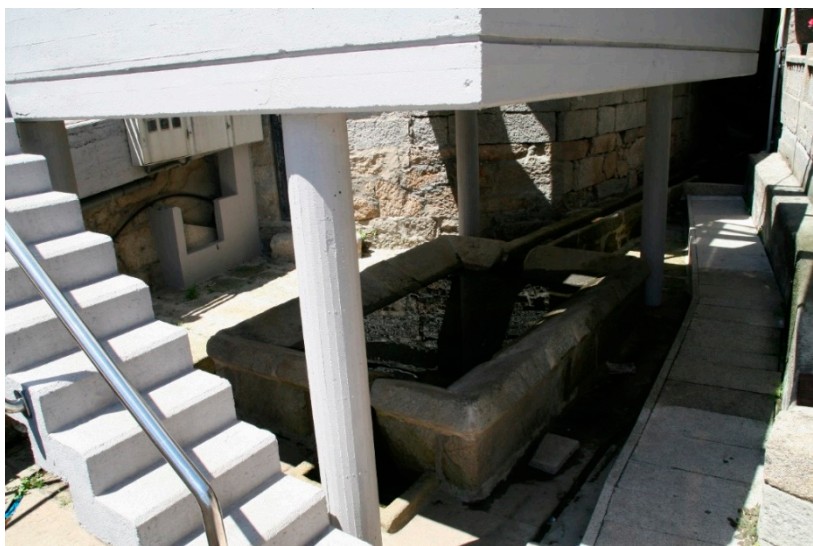

**Figure 4.** Wash tank on the first level of the community laundry, Combarro, designed by Pascuala Campos de Michelena (Authorship: AFC).

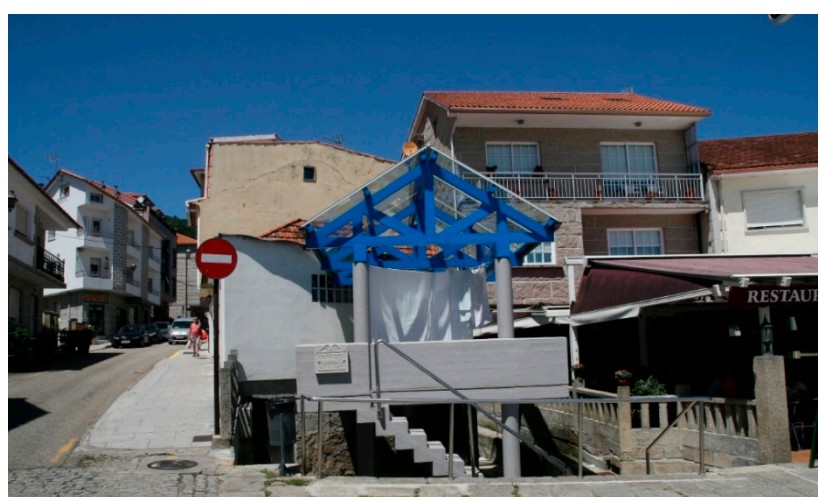

**Figure 5.** Plastic roof on the second level of the community laundry, Combarro, designed by Pascuala Campos de Michelena (Authorship: AFC).

Tzonis and Lefaivre (1985) interpret the volume as a reference to the traditional granaries that, in turn, resembles a classic temple. With this analogy, Pascuala Campos exposed a common and usually invisible female task and by doing so, she dignified it. Therefore, it can be interpreted as both a celebration of the local identity and a protest against the invisibility of women's work in these rural communities.

Although industrial materials were introduced in Pascuala Campos's plan for Combarro, as seen in the above example (Figure 5), traditional materials (such as stone and wood) prevailed on the surfaces' treatment.

If for Pascuala Campos the public space is the expression of the collective, then in Combarro it expresses a complex and diversified way of life (Campos de Michelena 1985). The plan she presented for Rualeira Square tried to embrace that collective spirit, without forgetting the individual needs and expectations. Her proposals enhanced the relationship between users and space, thus fitting in with the collective, but at the same time creating areas where the user could feel at ease or that invited them to explore the spaces and the streets. In short, they enhanced both interpersonal and spatial relationships.

4.2.2. Fishery Training School, Punta de Niño do Corvo, Arousa (1988)

Punta de Niño do Corvo, is situated in Arousa Island, in the center of *Ria de Arousa*. Traditionally occupied by a community of fishermen, the island was connected by a 2 km bridge to the mainland in 1985.

Three years later, Pascuala Campos won the architectonic competition for the design project of the island *Fishery Training School*. The project, developed between 1988 and 1990, revealed, once again, Pascula Campos's particular understanding of the essence of architectural design and its method of development.

The center is mainly devoted to aquaculture and diving practice, but also works as a research center. Its installations are one of the best equipped in all of Spain, and recently became a reference in these areas. It is equipped with several classrooms, a library, and an assembly hall, several areas devoted to aquaculture, laboratories, two swimming pools, and other facilities for diving practice.

In Pascuala Campos's own words, this project was inspired by the image of the old industrial spaces devoted to the conservation of fish, which was traditionally done by salting it in specific industrial facilities locally called *salazóns*. The *salazóns* were later abandoned as the conservation of fish was done in preserves, which were prepared in another typical Galician industrial facility, called *conserveiras*. The *salazóns* and the *conserveiras* constituted a landmark in Illa de Arousa, and thus were chosen by Pascuala Campos as the main inspiration for her *Fishery Training School* (Campos de Michelena 1992; Campos de Michelena 1993; Baldellou 1995), with its proximity to the shore and its integration with the rock-strewn environment, therefore showing her characteristic approach regarding the profound respect her work shows towards the identity of the place and its history.

Moreover, the integration with the surroundings is complemented with the abolishment of the interior/exterior boundaries. The interior spaces are distributed and oriented in order to take advantage of the views (Campos de Michelena 1992, 1993), thus responding to her idea of private spaces that are no longer enclosed and segregated.

Her project also shows a particular attention to the specific needs of the program and a more profound understanding of its meaning, in her belief that "a building dedicated to teaching must offer the first lesson of the day" (Campos de Michelena 1992, p. 47).

The program resolution is very simple in its form and, at the same time, complex in its multiple associations and meanings. The entire building is designed around two inner courtyards (Figure 6), as a reminiscence of a cloister, a place of quiet, reflection, and knowledge, traditionally associated with learning facilities (AV 1993).

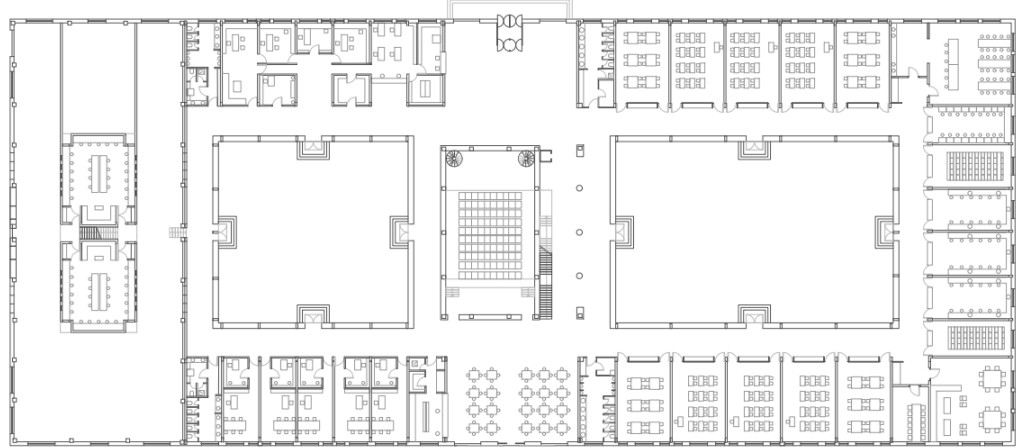

**Figure 6.** Plan of the Fishery Training School, Punta de Niño do Corvo, Arousa, designed by Pascuala Campos de Michelena (Authorship: AFC).

The main volume, in the intersection of the two courtyards, contains the main public spaces: the entrance and the assembly hall on the ground level, the gym and the swimming pool in the basement, and the library on the first floor.

A big staircase gives access to the library, while its interior marks the separation between its two spaces: electronic library and reading room. This is clearly one of the most privileged spaces within the building, with its volumetry accentuated in the composition and its proportion that can be interpreted as a reference to a granary (AV 1993).

On one side of this central volume is the cafeteria, which, as a place of meeting, enjoyment, and human relations, is carefully designed as one more classroom (Campos de Michelena 1992).

Placed on the other side are the research facilities, designed as another compact volume, with the fish tanks in the basement, and the laboratories and classrooms above it, both connected by a walkway that allows the observation of the fish tanks (Campos de Michelena 1992). Around the second courtyard are the remaining classrooms, and the administrative area.

The overall composition is characterized by a low elongated volume that marks a horizontal line with its powerfull stone walls, only interrupted by the roof slope of the research facilities volume on one side, and the clear and detached volume of the library in the center (Figure 7) which projects, in contrast, a clear vertical line (AV 1993).

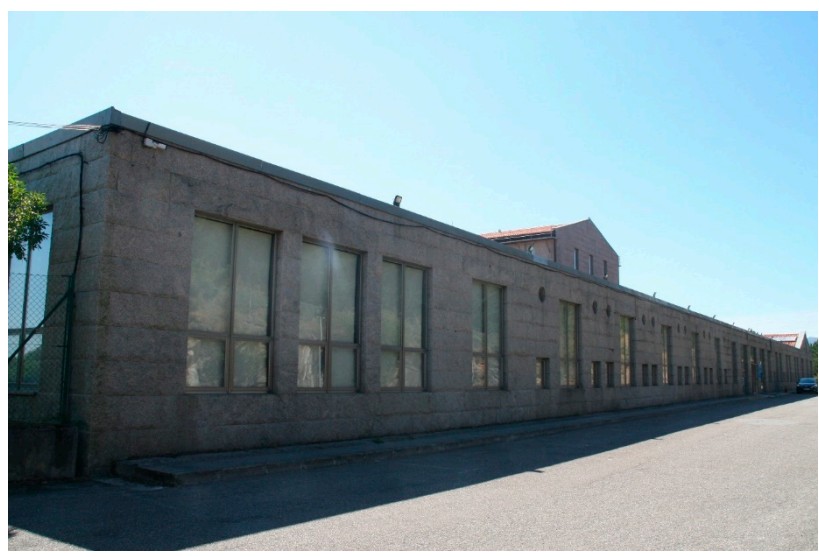

**Figure 7.** General view of the Fishery Training School, Punta de Niño do Corvo, Arousa, designed by Pascuala Campos de Michelena (Authorship: AFC).

The gym and the swimming pool have their independent entrance, which was devised as a way of allowing the local residents to have access to these facilities, hoping that they would take advantage of swimming classes, in order to avoid the constant drowning accidents associated with fishing activities (Campos de Michelena 1992). This concern is, once again, an argument towards Pacuala Campos's ability to satisfy the specific needs of the academic community, without forgetting the collective needs of the fishermen and their families.

## 5. Discussion

Pascuala Campos believes that architecture is always a process of reflection, in which intuition plays a decisive role. Only through her/his own emotions and intuitions can the architect find a solution for an inclusive urban or architectural design.

In Combarro, this sensibility of the architect can be observed in the way she reads interconnections between agriculture and the fishing culture, two realities that she tried to integrate. And it reappears in the Fishery Training School with a different perspective over the Galician traditional fishing culture,

this time with the reference to the *salgadeiras* and *conserveiras* that once marked the image of Arousa Island, and the image of the granary is also recovered in the library volume. These are solutions that allow her to recover the forgotten heritage (past) and give it a new utility (iconic elements), so that the history of the place (historical present) is not lost.

Another clear perspective that may have great relevance in the field of architecture concerns spaces that are individually envisioned, focusing on the user, but these individual needs and expectations must be integrated into the "city" as a whole, and thus must respond to the needs of the collective. In Combarro, one can find traces of this perspective in the way she envisioned spaces such as the communitarian laundry, or Rualeira Square, as spaces for encounters and relations. It is then once more detectable in her criticism towards the exclusion of community participation during the classification of Combarro as an Artistic Historic Site. On the contrary, she makes an effort to simultaneously integrate the collective and individual needs. This goal is evident in the solutions she proposed for Combarro, and in the way she designed them.

In her urban intervention in Combarro, she valued the collective without distinction or hierarchy. It established the recreation of spaces already built and the construction of new spaces of neutrality, considering the identity of the place, the transgression (understood as the breaking of limits), and the relations of the community with space itself.

Moreoover, in the Fishery Training School, she starts by taking into consideration the specific needs of the research/training program, but ends up broadening her perspective to include the need of the collective, opening the building to the Arousa fishing community.

Pascuala Campos's designs reflect her perspective of life, materialized through spaces of passage, meeting, relationships, emotions, and intimacy, always considering the community as a whole, without forgetting the individual needs of its inhabitants.

One can discuss that some part of this attitude can be attributed to the principles defended by Galician critical regionalism, this being the case of her respect to the place, its history, and its traditions. Others can only be understood from the viewpoint of an architecture built with a gender perspective. The work method that values intuition and personal feelings is a good example of this new dynamic, but also the appreciation of spaces of communication and intimacy, a framework aligned with the ideas defended by Karen A. Franck. Pascuala Campos's intervention in the community laundry at Combarro speaks for itself, but it can only be truly understood through her very personal understanding of space, gender, and community.

## 6. Conclusions

This theoretical reflection intended to collaborate a well-deserved tribute to Pascuala Campos de Michelena, trying to frame her contribution to the discipline, based on her theoretical and professional production.

The first conclusion is that her contribution can be seen from the very beginning of her career, although at first, her merit was not fully recognized due to her feminine condition. With strength and perseverance, she was able to make her voice heard and earn the recognition her work deserved.

So the next big question is: How was this possible in a context of profound discrimanation towards women?

Ironically, the answer is also intimately related to her very acute sence of what it means to be a woman, in other words, with her own feminine identity. Pascuala Campos had the ability to transform her biggest obstacule into her greatest strength.

One can perceive how Pascuala Campos integrates, both in her professional practice and her academic work, a new dynamic of action based on gender perspective that may be seen as one of the distinctive features of her built work. This characteristic, although it is not yet duly recognized, may well be the factor that granted her work some recognition in the Galician architectonic context. In the interpretation of the authors of the present paper, this what makes it unique. Her female special

sensitivity to the needs of the most fragile shape her "new" perception of space, and her militancy as a feminist activist gives her the strength not to abdicate her identity.

From all perspectives—personal, civic, theoretical, and professional—she cracked the rules that dictated how women should act and think. Her example inspired new generations of female architects to actively claim the "recognition and visibility" of women in architecture.

Consequently, she helped in creating a multicolored and diverse architecture, built through communicating spaces, envisioned as places of meetings, relationships, emotions, and intimacy, in trying to build a plural and diverse reality.

**Author Contributions:** Conceptualization, A.F.C. and G.S.; Formal analysis, A.F.C.; Investigation, A.F.C.; Methodology, A.F.C. and G.S.; Supervision, G.S., P.G. and M.C.; Writing—original draft, A.F.C. and G.S.; Writing—review & editing, A.F.C. and G.S. All authors have read and agreed to the published version of the manuscript.

**Funding:** This research received no external funding.

**Conflicts of Interest:** The authors declare no conflict of interest.

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
