# Peer review of "The Contribution of the Architect Pascuala Campos to the Implementation of a Gender Perspective in the Galician Context"

_arts, 1998_

Round 1

Reviewer 1 Report

This article aims to present a historical view of Pascuala Campos’s role in Spanish architecture of the 20th century, highlighting her special contribution  to advancing the position of women in Spanish architecture.

The author seems to ask a historical question which could best be examined through examination of appropriate sources - archival research, and/or interviews. Alternatively, the author could frame this as a theoretical question and present a theoretical analysis, in which writings or built works are analyzed. In the discussion of methodology, the author seems to have difficulties with the terms presented. The author mentions case studies and qualitative methods, which are not used in the study.

The article is weakly sourced and needs additional citations. For example factual information in lines 35-42 needs to be cited. Lines 43-44 mention “a large variety of literature” yet the author never specifies what literature. In lines 45-72 the author presents information that remains unverified, because no source is given. This pattern continues throughout the article. A historical examination must be carefully sourced and theoretical discussions must build upon the work of others!

The author does not seem to be familiar with the vast theoretical literature on gender and architecture - at least none of the pertinent literature is mentioned anywhere in the article. To frame the study, it would be helpful to discuss theoretical concepts that are informing the work.

The main sources for this study seem to be secondary literature (published books or papers). The article should strive to make new contributions to the field by analyzing primary sources.

In the analysis of Campos’s projects, the author does provide some interesting insights. If the author were to expand upon this discussion and link it to a well-developed feminist-theoretical discussion, the article would achieve its aim of presenting Capos’s work through a gender-theoretical lens. The author’s presentation of how the Campos’s life and work and gender questions are linked is weak at best. After reading the article, the reader is left with the impression that Campos somehow achieved success as an architect (despite being a woman), but we never really learn why or how. The conclusion begins to hint at some of the reasons, but these should be better developed in the body of the text.

The article needs to be language-edited for better clarity. Some of the sentences use wording or syntax that does not conform to standard English usage.

Author Response

Taking into consideration the review report available the research paper “The Contribution of the Architect Pascuala Campos to the Implementation of a Gender Perspective in the Galician Context” was significantly revised. As detailed below:

  1. Taking into consideration your suggestion the paper was framed as a theoretical analysis since it is based on the analysis of Pascuala Campos` theoretical and built work.
  2. In the discussion of methodology, the mentions to case studies and qualitative methods were eliminated (cf. abstract and line 116).
  3. Additional citations were provided, revealing the sources for the information present on the text (cf. the references introduced, for example, in lines 64-68 and 69-73).
  4. The theoretical concepts that set de background for this article were introduced at the beginning of the text (lines 22-49).
  5. In the impossibility to, at this moment to present first-hand sources, we tried to develop a link between the theoretical feminist background and our analysis of Pascuala Campos Work (cf. lines 402-460).
  6. These revisions tried to strengthen our perceptions of how and why Pascuala Campos achieved success as an architect in a context discriminatory towards women.

English language and style was revised trying to improve the clarity of the text.

We hope to have answered in the best possible way to your insights and revision comments.

Best regards

The authors.

Reviewer 2 Report

1 very good paper. page 2, third issue (lines 64-72) should make it clear how, and what the vision of space is and how it is used.  While it may be new way of expressing an equitable vision and a unique gender perspective,  is this a "method" / way of working ?  Does it result in a different kind of design? different type of space? Organization? use of materials? work with clients? etc.  

Author used the Intervention in Comburro to illuminate Campos's ideas related to the construction of space, intuition, etc.  This could be expanded with specific examples within her own projects here and to specifically address these ideas in more detail..  I understand that the village itself was an example of a number of these ideas, it would be very helpful to see how she developed the idea within the project itself or within another built design project. 

2 more images that demonstrate the assertions and examples.  Figure 4.1 is impossible to understand in relation to the text.

3. Grammar and some misuse of english words -- switches between past tense and present tense. I prefer past tense when speaking and writing about what was done, developed, etc in the past. Some typos, misspellings, misuse of articles and words.  Author should have another editor review the paper.

4. Line 298 -- replace "him" with "her".

5. in Conclusions section 6: lines 340-346.  Examples, specific examples, that support your conclusions here should appear in the text.  

Author Response

Taking into consideration the review report available the research paper “The Contribution of the Architect Pascuala Campos to the Implementation of a Gender Perspective in the Galician Context” was significantly revised. As detailed below:

  1. Lines 64-72 were completely rewritten in order to make it clear how and what is this new vision of space. Furthermore, additional insights to these issues were introduced in the text (cf. for example lines 38-49).
  2. We discarded the analysis of other built works from Pascuala Campos, because, the most part of her work was done in collaboration, first with her husband and later with other architects. Which constituted a problem concerning authorship, making it difficult to highlight the personal contribution of Pascuala.
  3. New images were introduced in order to demonstrate assertions and examples. Meanwhile, although we agree that the scale of figure 4.1. (now 4.2.) can make it a little difficult to understand we decided to maintain the image as presented since the purpose was to present the general proposal for Combarro. We hope that in conjunction with the pictures we introduced it will be easier to read.
  4. In the conclusions, specific examples are provided as requested (cf. line 426-430).
  5. English language and style were revised trying to improve the clarity of the text.

We hope to have answered in the best possible way to your insights and comments,

Best regards

The authors.

Round 2

Reviewer 1 Report

This text has vastly improved since the first version! It is now very coherent in its argument and the citations are used appropriately.

I would make three very small changes. In line 343, omit “As discussed above”. It’s a weak start to any section. And in line 364 omit “In conclusion”. The conclusion section starts with the next paragraph. In line 423, omit "So" at the start of the sentence. This word is a filler, and weakens the sentence.

Otherwise, I have no quibbles and thoroughly enjoyed reading the piece!

Author Response

Dear reviewer,

Taking into consideration your report:

  1. In line 343 the expression “As discussed above” was omitted.
  2. In line 364 it was also omitted the expression “In conclusion”.
  3. Finally, the beginning of line 423 was also rewritten.

We sincerely thank you for your contribution,

Best Regards,

The Authors